# PROBABILITY-DEPENDENT GRADIENT DECAY IN LARGE MARGIN SOFTMAX

## ABSTRACT

In this paper, a gradient decay hyperparameter is introduced in Softmax to control the probability-dependent gradient decay rate. By following the theoretical analysis and empirical results, we find that the generalization and calibration depend significantly on the gradient decay rate as the confidence probability rises, i.e., the gradient decreases convexly or concavely as the sample probability increases. Moreover, optimization with the small gradient decay shows a curriculum learning sequence where hard samples are in the spotlight only after easy samples are convinced sufficiently, and well-separated samples gain a higher gradient to reduce intra-class distance. Unfortunately, the small gradient decay exacerbates model overconfidence, shedding light on the causes of the poor calibration observed in modern neural networks. Conversely, a large gradient decay significantly mitigates these issues, outperforming even the model employing post-calibration methods. Based on the analysis results, we can provide evidence that the large margin Softmax will affect the local Lipschitz constraint by regulating the probability-dependent gradient decay rate. This paper provides a new perspective and understanding of the relationship among large margin Softmax, curriculum learning and model calibration by analyzing the gradient decay rate. Besides, we propose a warm-up strategy to dynamically adjust gradient decay.

## 1 INTRODUCTION

Softmax function combined with a cross-entropy loss (CE) as the logically reasonable loss function in empirical risk minimization of classification, has been recognized as the state-of-the-art base objective function in practical neural network optimization. Softmax function converts the whole output space into an approximate probability distribution as a measure of the distance between the predicted distribution and the label. However, the modern models often demonstrate inadequate confidence calibration in probability distribution processed through Softmax mapping. Specifically, these probability outputs display unwarranted over-confidence (Guo et al., 2017). Furthermore, researchers have identified that achieving high accuracy in classifiers and calibrating the model confidence are distinct objectives (Wenger et al., 2020). This scenario emphasizes the necessity to study the calibration of model output uncertainties in optimization.

The Softmax function is usually defined by a single hyperparameter, the temperature $\tau$, which scales the smoothness between Softmax and max function. The temperature $\tau$ is often discussed in contractive learning (Wang & Liu, 2021), knowledge distilling (Hinton et al., 2015), natural language processing (Liu et al., 2021) and so on. In some specific tasks, a small preset temperature $\tau$ can produce a hefty penalty on hard negative samples to force more significant inter-class discrepancy. Moreover, the penalty distribution tends to be more uniform as the temperature increases (Guo et al., 2017). It seems reasonable that static model calibration plays the pivotal role (Zhang et al., 2018). Nevertheless, the literature (Agarwala et al., 2020) demonstrates that the dependence of the generalization on temperature is due to a dynamical phenomenon rather than model calibration.

Similarly, the hard mining strategy explicitly emphasizes more challenging samples by adjusting weights of easy samples or hard samples in Softmax variations (Wang et al., 2020; Ren et al., 2017). Mining-based Softmax concentrates on the informative samples, so can learn more discriminative features (Shrivastava et al., 2016; Huang et al., 2020). Selecting the value sample and removing noisy data are the technical foundation of the hard mining strategy. On the other hand, a soft mining

strategy, like focal loss, smooths the mining strategy by introducing a modifiable hyperparameter so that the hard sample can be given more importance (Zheng et al., 2021). Due to its broad applicability, it has become a prevailing loss (Lin et al., 2020).

Another branch of Softmax research is the large margin Softmax, which increases the feature margin from the perspective of the ground truth class. Liu et al. introduced Large-margin Softmax (L-Softmax) (Liu et al., 2016) and Angular Softmax (A-Softmax) (Liu et al., 2017) to impose discriminative constraints on a hypersphere manifold to encourage intra-class compactness and inter-class separability between learned features. Wang et al. proposed a more interpretable way to import the angular margin into Additive Margin Softmax (AM-Softmax) (Wang et al., 2018). In (Deng et al., 2019), the Additive Angular Margin Loss (ArcFace) showed a clear geometric interpretation due to its exact correspondence to geodesic distance on a hypersphere. Unlike the hard mining strategy, large margin Softmax not only swells the inter-class distance by adjusting the temperature but also remains focused on the intra-class distance. However, to the best of our knowledge, there is a lack of corresponding explanatory work from dynamic training performance. Our findings will demonstrate that the "margin" could contribute to miscalibration in modern models.

This paper introduces a hyperparameter $\beta$ in Softmax, which controls the probability-dependent gradient decay as the sample confidence probability rises. From the theoretical analysis, we can conclude that the smaller the hyperparameter $\beta$ is, the smoother the local $L$-constraint of the Softmax is. It means that the model with a smaller $\beta$ can obtain a rapider convergence rate in the initial phase. As shown in Fig. 1, minor gradient decay produces a higher gradient to the well-separated sample to shrink the intra-class distance at the expense of confidence of some hard negative samples due to the limited network capacity. The training with a slight gradient decay shows a similar curriculum learning idea (Bengio et al., 2009; Jiang et al., 2018; Zhou & Bilmes, 2018) that the hard samples will be optimized only after the easy samples have been convinced sufficiently. Unfortunately, small probabilistic gradient decays worsen the miscalibration of the

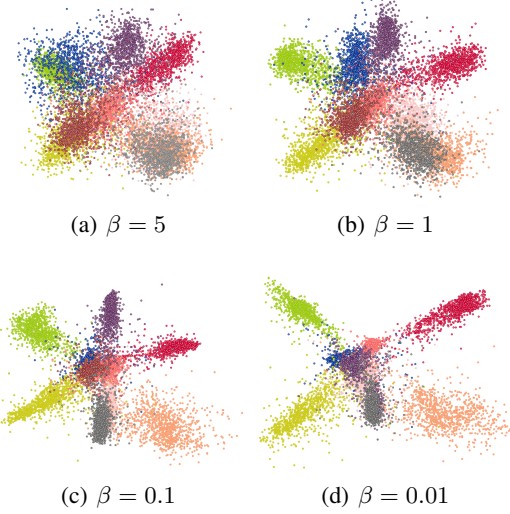

(a) $\beta = 5$      (b) $\beta = 1$

(c) $\beta = 0.1$      (d) $\beta = 0.01$

Figure 1: The features distribution with different gradient decay factors.

modern model, while larger decay rates generally smooth the training sequence, alleviating this issue. This paper analyzes the dynamic training phenomenon with different $\beta$ and provides a new understanding of large margin Softmax by considering the effect of the hyperparameters $\beta$ on gradient decay rate. Besides, we propose a warm-up training strategy to set an over-small initial hyperparameter $\beta$ to speed up the convergence rate. Then, $\beta$ is enlarged to an adequate value to prevent over-confidence.

## 2 PRELIMINARIES

**Parametric Softmax** Considering a dataset $\left\{(x^i, y^i)\right\}_{i=1}^{N} \subset \mathbf{R}^n \times \mathbf{R}^m$ and classifier $f$ maps $x$ to the outputs $z_j, j = 1, \ldots, m$ on $m$ classes. The associated confidence score of the predicted label in baseline is $\hat{p} = \max s_j(z), j = 1, \ldots, m$, where $s(\cdot)$ represents Softmax mapping $R^m \to R^m$. Softmax cross-entropy (CE) is expressed as

$$J = -\log \frac{e^{z_c}}{\sum_{j=1}^{m} e^{z_j}} \tag{1}$$

where $z_j, j = 1, \ldots, m$ represent the outputs of $m$ labels. $c$ represents the truth class in $m$ classes.

We introduce two hyperparameters in the Softmax mapping, which is expressed as follows:

$$J = -\log \frac{e^{z_c/\tau}}{\sum_{j \neq c} e^{z_j/\tau} + \beta e^{z_c/\tau}} \tag{2}$$

The parametric Softmax cross-entropy can be approximated as the following max function, as shown in (3). Minimizing this max function is expected that output $z_c$ can be larger than other class outputs $z_j, j = 1, \ldots, m, j \neq c$, which is in line with the logic of the one-versus-all classification decision-making $cls(z(x)) = \max\{z_j(x)\}, j = 1, \ldots, m$.

$$\lim_{\tau \to 0} -\log \frac{e^{z_c/\tau}}{\sum_{j \neq c} e^{z_j/\tau} + \beta e^{z_c/\tau}} = \lim_{\tau \to 0} \max\{\log \beta, z_j - z_c/\tau, j = 1, \ldots, m, j \neq c\} \quad (3)$$

**Temeperature and Model Calibration** Perfect calibration of neural network can be realized when the confidence score reflects the real probability that the classification is classified correctly. Formally, the perfectly calibrated network satisfied $P(\hat{y} = y | \hat{p} = p) = p$ for all $p \in [0, 1]$. In practical applications, the sample is divided into $M$ bins $\{D_b\}_{b=1}^M$. According to their confidence scores and the calibration error, an approximation is calculated for each bins $\{D_b\}_{b=1}^M$. $D_b$ contains all sample with $\hat{p} \in \left[\frac{b}{M}, \frac{b+1}{M}\right)$. Average confidence is computed as $conf(D_b) = \frac{1}{|D_b|} \sum_{i \in D_b} \hat{p}^i$ and the bin accuracy is computed as $acc(D_b) = \frac{1}{|D_b|} \sum_{i \in D_b} I(y_c^i = \hat{y}_c^i)$. Expected Calibration Error (ECE) (Naeini et al., 2015) is calculated as follows.

$$ECE = \sum_{b=1}^M \frac{|D_b|}{N} |acc(D_b) - conf(D_b)| \quad (4)$$

Modern models present overconfidence in the estimation of output uncertainty (Guo et al., 2017). Temperature scaling is the most popular post-processing calibration method (Krishnan & Tickoo, 2020; Karandikar et al., 2021) by adjusting temperature $\tau$. In model training, some papers also attribute the improvement of the temperature scaling solely to the calibration of the model confidence (Wang & Liu, 2021). However, a challenge was raised in (Agarwala et al., 2020) that the dependence of the generalization on temperature is due to a dynamical phenomenon rather than model confidence.

**Soft Margin** In (3), the hyperparameter $\beta$ represents the soft margin in decision space. So the cross-entropy itself can be interpreted as a margin-based loss (Zheng et al., 2021). However, owing to the distance distortion between input and representation spaces, the large margin in the input space of models is not maximized simultaneously by large margin Softmax. That is reflected that a more considerable margin does not mean better generalization. Besides, the margins defined based on different criteria realize different performances, i.e., angular margin (Deng et al., 2019) or cosine margin (Liu et al., 2016). So the interpretation of its effect is slightly ambiguous.

On the other hand, the model training is associated with $J_i, i = 1, \ldots, N$, which $N$ represents sample number, and $N$ optimization problems are combined as a multi-objective optimization $\sum_{i=1}^N J_i$. The coupling effect among the samples should be considered in the optimization.

## 3 GRADIENT DECAY

### 3.1 GRADIENT DECAY HYPERPARAMETER

We consider the Softmax with the sole hyperparameter $\beta$. The temperature $\tau$ is set to 1.

$$J = -\log \frac{e^{z_c}}{\sum_{j \neq c} e^{z_j} + \beta e^{z_c}} \quad (5)$$

Let us first consider the gradient of the Softmax.

$$\frac{\partial J}{\partial z_c} = -\frac{\sum e^{z_j} - e^{z_c}}{\sum e^{z_j} + (\beta - 1) e^{z_c}} \quad (6)$$

$$\frac{\partial J}{\partial z_j} = \frac{e^{z_j}}{\sum e^{z_j} + (\beta - 1) e^{z_c}} \quad (7)$$

We introduce probabilistic output $p_j = \frac{e^{z_j}}{e^{z_1} + \cdots + e^{z_m}}$ as an intermediate variable. Then we obtain:

$$\frac{\partial J}{\partial z_j} = \begin{cases} -\dfrac{1 - p_c}{1 + (\beta - 1)p_c}, & j = c \\[3mm] \dfrac{p_j}{1 + (\beta - 1)p_c}, & j \neq c \end{cases} \quad (8)$$

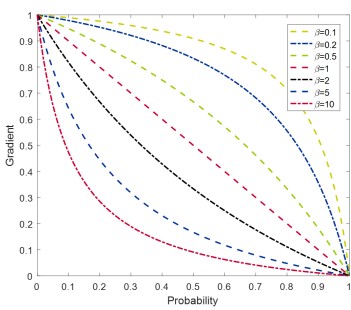

Figure 2: The gradient magnitude.

Since $\left|\frac{\partial J}{\partial z_c}\right| + \sum_{j \neq c}\left|\frac{\partial J}{\partial z_j}\right| = 2\left|\frac{\partial J}{\partial z_c}\right|$, $\left|\frac{\partial J}{\partial z_c}\right|$ can represent the gradient magnitude of this sample. Moreover, $p_c$ can represent the confidence of the model for this sample. In (8), we can conclude that $\beta$ determines the gradient magnitude related to sample probability confidence. We define the gradient magnitude $G = -\frac{\partial J}{\partial z_c}$. Then, we can get first-order and second-order derivatives of $G$ with respect to $p_c$.

$$\frac{\partial G}{\partial p_c} = \frac{-\beta}{(1 + (\beta - 1)p_c)^2} \tag{9}$$

$$\frac{\partial^2 G}{\partial p_c{}^2} = \frac{2\beta(\beta - 1)}{(1 + (\beta - 1)p_c)^3} \tag{10}$$

When $\beta > 1$, $\frac{\partial^2 G}{\partial p_c{}^2} > 0$, gradient magnitude decreases concave monotonically as the sample probability rises; When $1 > \beta > 0$, $\frac{\partial^2 G}{\partial p_c{}^2} < 0$, gradient magnitude shows convex monotonically decreasing as the sample probability rises. As shown in Fig. 2, $\beta$ controls the gradient decay rate as the sample probability rises. The smaller hyperparameter $\beta$ shows a lower gradient decay rate in the initial phase. Furthermore, the gradient magnitude decays rapidly after the probability exceeds a certain value, which can be interpreted as a soft probability margin.

However, derivatives of $G$ to $p_c$ seem to be abstract. So, we need to obtain the second-order and the third-order derivatives of $J$ to the truth class output $z_c$. We introduce the intermediate variable $p_c$.

$$\frac{\partial^2 J}{\partial z_c{}^2} = \frac{\partial^2 J}{\partial z_c \partial p_c}\frac{\partial p_c}{\partial z_c} \tag{11}$$

Because $\frac{\partial p_c}{\partial z_c} = p_c(1 - p_c)$. So we get $\frac{\partial^2 J}{\partial z_c{}^2} = \frac{\beta p_c(1-p_c)}{(1+(\beta-1)p_c)^2}$ and

$$\frac{\partial^3 J}{\partial z_c{}^3} = \frac{\beta p_c(1 - p_c)}{(1 + (\beta - 1)p_c)^3}(1 - (1 + \beta)p_c) \tag{12}$$

$\frac{\beta p_c(1-p_c)}{(1+(\beta-1)p_c)^3} > 0$ is constant since $\beta > 0$ and $1 > p_c > 0$. So $\frac{\partial^3 J}{\partial z_c{}^3} < 0$ when $p_c > \frac{1}{1+\beta}$ ; $\frac{\partial^3 J}{\partial z_c{}^3} > 0$ when $p_c < \frac{1}{1+\beta}$. We concentrate on the change of the gradient magnitude $G$. Thus, the magnitude shows convex monotonically decreasing as $z_c$ increases when $p_c < \frac{1}{1+\beta}$, and concave monotonically decreasing when $p_c > \frac{1}{1+\beta}$. $p_c = \frac{1}{1+\beta}$ is the inflection point of gradient as the $z_c$ increases. $\beta$ determines the inflection point.

The magnitude of the gradient always decays from 1 to 0. As shown in Fig. 2, smaller $\beta$ produces a smoother decay in the initial phase, which results in a larger magnitude in the whole training. The inflection gradually moves away from the initial point $z_c = 0$ so that a smooth gradient and large magnitude can dominate training, as shown in Fig. 7. So, a small hyperparameter $\beta$ induces a low gradient decay rate and large gradient magnitude.

Let us consider two extreme cases: $\beta \to 0^+$ and $\beta \to +\infty$.

$$\lim_{\beta \to 0^+} G = \lim_{\beta \to 0^+} \frac{1 - p_c}{1 + (\beta - 1)p_c} = 1 \tag{13}$$

$$\lim_{\beta \to +\infty} G = \lim_{\beta \to +\infty} \frac{1 - p_c}{1 + (\beta - 1)p_c} = 0 \tag{14}$$

Obviously, $\beta \to 0^+$ will keep the sum of the gradient amplitudes $\left|\frac{\partial J}{\partial z_c}\right| + \sum_{j \neq c}\left|\frac{\partial J}{\partial z_j}\right|$ unchanged. In Fig. 2, the curve will be approximated as a step function where $G = 1, p_c < 1$ and $G = 0, p_c = 1$. On the other hand, $\beta \to +\infty$ forces the gradient down rapidly to 0. It is reflected in the changes of the convexity of the curves and the panning of the inflection point.

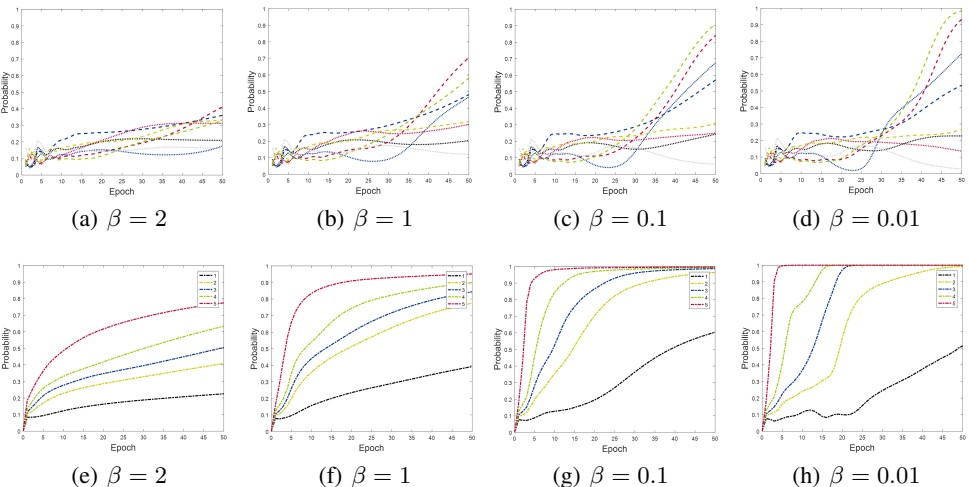

(a) $\beta = 2$      (b) $\beta = 1$      (c) $\beta = 0.1$      (d) $\beta = 0.01$

(e) $\beta = 2$      (f) $\beta = 1$      (g) $\beta = 0.1$      (h) $\beta = 0.01$

Figure 3: Dynamic confidence during training of three-layer FCNN on MNIST. **Top row a-d: Confidence of some representative samples.** The small gradient decay rate improves the probability output $\hat{p}$ of some samples while some samples receive even less probability $\hat{p}$, such as gray curve. **Bottom row e-h: Mean confidence of five groups of samples.** All the samples are divided into five groups "1-5" from low to high according to posterior probability $\hat{p}$. When $\beta$ decreases to 0.1, the average probability of all groups is raised. However, when $\beta$ continues to decrease to 0.01, the average probability of the group with the smallest probability becomes progressively smaller.

## 3.2 HOW DOES THE GRADIENT DECAY RATE AFFECT THE MODEL PERFORMANCE?

Fast gradient decay produces a slight gradient in the early phase of optimization. So the training may be prone to trap into local minima before sufficient learning in the early training phase. From the perspective of dynamic multi-objective optimization $\sum_{i=1}^{N} J_i$, hard objectives with smaller posterior probability $\hat{p}$, can always stay consistent with the easy objectives with larger posterior probability $\hat{p}$ in optimization due to the large probability-dependent gradient decay, as shown in Fig. 3. The gradient of well-separated sample is forced to decrease rapidly. A more significant constraint exists on the synchronization of the optimization process for different samples. Consequently, there is a lack of distinction between samples, potentially leading to model under-confidence.

A gradual decrease in gradient magnitude consistently prioritizes well-separated samples during training. The influence of easy samples persists until reaching the predefined margin. Consequently, well-separated samples receive ample attention and output feature of those samples can be gathered more compactly, as shown in Fig. 1. Then the intra-class variance can be shrunk and some discriminative features can be learned by the higher gradient (Ranjan et al., 2017).

Furthermore, the low gradient decay training strategy is similar to curriculum learning. That is, the samples should be learned strictly sequentially from easy to hard. In curriculum learning, the samples usually are labeled as "easy" or "hard" by complicated offline or online strategies (Tang & Huang, 2019). A smooth gradient implicitly ensures the strict training sequence and softly divides the sample optimization by posterior probability $\hat{p}$. As shown in Fig. 3, the optimization of Softmax with $\beta = 1$ keeps relatively consistent, while the smaller $\beta$ shows distinguishability over different samples. In Figs. 3e-h, the groups "1-5" represent the five sample groups from "hard" to "easy" according to $\hat{p}$. The Softmax with smaller gradient decay is more inclined to mine more information in easy samples under a soft curriculum idea. It can be inferred that the smaller gradient decay can realize the stricter curriculum sequence. Moreover, small gradient decay usually performs better convergence rate, but too low gradient decay does not imply better results. As shown in section 4.2, small gradient decay may result in worse miscalibration, while larger decay rates alleviate this issue.

Gradually reducing the gradient decay facilitates effective learning of discriminative features from easy samples. From the comparison between Fig. 3f and Fig. 3g, the probability can be enlarged for all groups. The slight boost is very important to improve generalization and robustness, although the error is very small (Soudry et al., 2018). However, the network capacity is limited and there is no

Table 1: Confidence distribution of the samples with different gradient decay of three-layer FCNN on MNIST on MNIST. # indicates the number of samples that belong to the confidence interval.

| Gradient decay factor $\beta$ | 1 | 0.5 | 0.1 | 0.01 | 0.001 |
|---|---|---|---|---|---|
| $\#p_c \leq 0.2$ | 903 | 828 | 1105 | 1325 | 2375 |
| $\#0.2 < p_c \leq 0.4$ | 454 | 206 | 119 | 91 | 142 |
| $\#0.4 < p_c \leq 0.6$ | 528 | 245 | 132 | 92 | 116 |
| $\#0.6 < p_c \leq 0.8$ | 1291 | 484 | 191 | 100 | 193 |
| $\#0.8 < p_c \leq 1$ | 56824 | 58237 | 58453 | 58392 | 57174 |

free lunch (Ho & Pepyne, 2002). If we make excessive demands on the margin, some challenging samples always remain low confidence $\hat{p}$ according to the stringent curriculum learning sequence, as shown in grey curve of Fig. 3c and Fig. 3d. The result in Tab. 1 shows that the number of the samples with low confidence $\#p_c \leq 0.2$ increases as $\beta$ is set to an over-small value. The over-large margin in (Wang et al., 2018) will make some challenging negative samples under-confident under limited model capacity, since the model gives more priority to easy samples. The intra-class distance of the partial positive easy samples will be reduced at the expense of the inter-class distance of some hard negative samples near the decision boundary. So there is a clear difference between the large margin Softmax and hard mining strategies (Wang et al., 2020): the former focuses on the overall confidence or even more on the mining of easy samples while the latter focuses more on challenging samples.

**Confidence calibration between $\tau$ and $\beta$ :** The Softmax with small $\tau$ disperses the inter-class distance by adjusting the probability output to focus more on hard negative samples. Nevertheless, large $\tau$ can only smooth the output of all categories and cannot mine more information from simple positive samples. On the contrary, small $\beta$ makes the gradient decay slowly so that easy positive samples can be sufficiently learned up to high confidence as shown in Fig. 3. An appropriate $\beta$ can mining more discriminative features on the whole. Similarly, large $\beta$ only keeps the consistency of the overall sample training and cannot extract more meaningful features from challenging samples.

In terms of the training process, $\tau$ changes the probability distribution of the class outputs and $\beta$ determines the gradient magnitude assigned by the probability of belonging to the truth class. They improved the mining capability of Softmax in two different dimensions. So it is convinced that Softmax cross-entropy should be characterized with these two hyperparameters $\tau$ and $\beta$.

## 3.3 LOCAL LIPSCHITZ CONSTRAINT

Assume that the gradient of the function $J(z)$ satisfies Lipschitz constraint ($L$-constraint) that is

$$\|\nabla_z J(z + \Delta z) - \nabla_z J(z)\|_2 \leq L\|\Delta z\|_2 \tag{15}$$

For a second-order differentiable objective function, the above condition is equivalent to $\left\|\nabla_z^2 J(z)\right\|_2 \leq L$, where $L$ represents the degree of fluctuation of the gradient. Then we have the following inequality (Zhang et al., 2019).

$$J(z + \Delta z) \leq J(z) + \langle \nabla_z J(z), \Delta z \rangle + \frac{1}{2}L\|\Delta z\|_2^2 \tag{16}$$

The gradient descent is applied to optimization. $\Delta z = -\eta \nabla_z J(z)$, where $\eta > 0$ is the learning rate. Substituting it into (16), we obtain

$$J(z + \Delta z) \leq J(z) + \left(\frac{1}{2}L\eta^2 - \eta\right)\|\nabla_z J(z)\|_2^2 \tag{17}$$

So, $\frac{1}{2}L\eta^2 - \eta < 0$ is the sufficient condition for loss decline at each iteration. And $\frac{1}{2}L\eta^2 - \eta$ is the minimum value when $\eta^* = \frac{1}{L}$. The larger the magnitude of the gradient $\|\nabla_z J(z)\|_2^2$ is, the smaller the $L$-constraint is. Furthermore, the smaller $L$-constraint results in the rapider convergence (Zhang et al., 2019). The learning rate $\eta$ can be adaptively designed to maximize the convergence speed (Carmon et al., 2018). Unfortunately, $L$-constraint is an intrinsic property of the loss function.

Since $\left|\frac{\partial J}{\partial z_c}\right| = \sum_{j \neq c} \left|\frac{\partial J}{\partial z_j}\right|$ , we consider the only variable $z_c$ in Softmax. For function $J(z_c)$ as shown in (9-12), we obtain that the $\max \left\|\nabla_{z_c}^2 J(z_c)\right\|_2$ is $\frac{1}{4}$ when $p_c = \frac{1}{1+\beta}$. So $\beta$ cannot change the global $L$-constrain since it is a constant. However, the local $L$-constrain can be adjusted by overall panning. Thus, we can narrow $\beta$ and change the inflection point of the gradient $p_c = \frac{1}{1+\beta}$ so that the constant maximum is far from the initial point, allowing a larger range of the smooth gradient decay to occupy the optimization process. For example, we consider the local range $p_c \in [0, 0.5]$ and can obtain a local $L$-constrain of $\left\|\nabla_{z_c}^2 J(z_c)\right\|_2$ as follows:

$$
\begin{cases}
\left\|\nabla_{z_c}^2 J(z_c)\right\|_2 \leq \dfrac{\beta}{(\beta+1)^2}, \beta < 1 \\
\left\|\nabla_{z_c}^2 J(z_c)\right\|_2 \leq \dfrac{1}{4} \qquad\quad , \beta \geq 1
\end{cases}
\tag{18}
$$

So it can be demonstrated that the $\beta$ smaller is, $L$-constrain in the early phase of optimization smaller is. When $\beta$ is set to a small value, the learning rate $\eta$ of gradient descent in (18) can be amplified to accelerate the optimization. On the other hand, the gradient magnitude $\left\|\nabla_{z_c} J(z_c)\right\|_2^2$ of smaller $\beta$ is always greater than that of larger $\beta$. Besides, it is meaningful that we can change $\beta$ to control the local $L$-constraint of the Softmax loss in optimization. Some literature (Elsayed et al., 2018) have shown that the Lipschitz constraint of the gradient has been strongly related to model generalization and robustness. The regularization of the gradient $L$-constraint has been applied to obtain the large margin decision boundary or guarantee the stable optimization in Generative Adversarial Networks (GAN) (Jolicoeur-Martineau & Mitliagkas, 2019).

### 3.4 WARM-UP STRATEGY

Based on the above analysis, we can conclude that the smaller $\beta$ produces a larger gradient magnitude with less gradient decay in the initial phase and can realize faster convergence. However, some challenging samples may remain low confidence by over-small $\beta$. The risk of model overconfidence is exacerbated. Thus, we propose a warm-up training strategy, where over-small $\beta$ in the initial phase provides fast convergence with the smooth local $L$-constrain and increases gradually until an adequate set. Then, easy sample can be learned sufficiently in early training and over-miscalibration of final model can be avoid. In this paper, we use a simple linear warm-up strategy as $\beta = \frac{\beta_{end} - \beta_{initial}}{t_{warm}} t + \beta_{initial}$, where $\beta_{initial}$ and $\beta_{end}$ are preset initial and final values. $t_{warm}$ and $t$ represent the end iteration of warm-up strategy and current iteration, respectively. $\beta_{initial}$ gradually increases to preset $\beta_{end}$ in the training.

## 4 EMPIRICAL VERIFICATION

### 4.1 CONVERGENCE RATE AND ACCURACY

It is shown in Fig. 4 that the performance of different gradient decays on three-layer FCNN shows different phenomena. The Softmax with less gradient decay displays faster convergence and better generalization with discriminate feature learning. $\beta = 0.01$ realizes the better accuracy than traditional CE with $\beta = 1$ and the large gradient decay has worse performance .

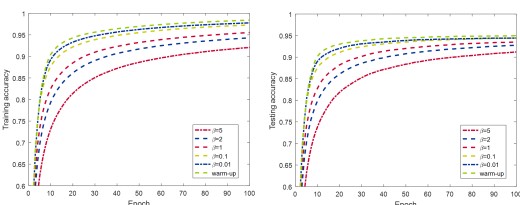

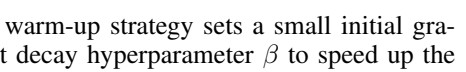

Figure 4: The performance on MNIST.

The warm-up strategy sets a small initial gradient decay hyperparameter $\beta$ to speed up the convergence rate and guarantees final stable performance by increasing $\beta$ to prevent over-confidence for easy sample. As a result, warm-up strategy achieves higher accuracy compared to a fixed gradient decay coefficient. Furthermore, decline curves of cross-entropy loss with different $\beta$ are shown in Fig. 5 to empirically show the high convergence rate of small gradient decay. It can be inferred that minor gradient decay shows the rapider convergence rate, although the gap decreases due to the

Table 2: Top-1 accuracy with different methods and gradient decay factors. The best results are in bold. Results are averaged over five runs with different seeds.

| Dataset | Model | Gradient decay factor $\beta$ | | | | | Warm-up | A-Softmax | Center loss |
|---|---|---|---|---|---|---|---|---|---|
| | | 5 | 2 | 1 | 0.1 | 0.01 | | | |
| SVHN | ResNet18 | $94.4_{\pm0.42}$ | $94.9_{\pm0.45}$ | $95.0_{\pm0.44}$ | $95.9_{\pm0.46}$ | $95.8_{\pm0.51}$ | $\mathbf{96.2}_{\pm0.41}$ | $95.9_{\pm0.42}$ | $95.8_{\pm0.45}$ |
| SVHN | ResNet35 | $95.1_{\pm0.40}$ | $95.3_{\pm0.43}$ | $95.5_{\pm0.46}$ | $96.2_{\pm0.46}$ | $96.1_{\pm0.45}$ | $\mathbf{96.3}_{\pm0.37}$ | $96.0_{\pm0.50}$ | $95.9_{\pm0.40}$ |
| SVHN | VGG16 | $95.0_{\pm0.51}$ | $95.1_{\pm0.56}$ | $95.3_{\pm0.59}$ | $\mathbf{96.2}_{\pm0.57}$ | $95.7_{\pm0.59}$ | $96.1_{\pm0.51}$ | $96.0_{\pm0.60}$ | $95.9_{\pm0.59}$ |
| SVHN | MobileNetV1 | $94.3_{\pm0.47}$ | $94.7_{\pm0.55}$ | $95.1_{\pm0.51}$ | $95.3_{\pm0.56}$ | $95.7_{\pm0.56}$ | $96.2_{\pm0.50}$ | $\mathbf{96.3}_{\pm0.52}$ | $95.8_{\pm0.49}$ |
| CIFAR-10 | ResNet18 | $94.3_{\pm0.31}$ | $94.8_{\pm0.35}$ | $94.9_{\pm0.39}$ | $94.9_{\pm0.39}$ | $95.2_{\pm0.41}$ | $\mathbf{95.5}_{\pm0.37}$ | $95.1_{\pm0.42}$ | $94.8_{\pm0.41}$ |
| CIFAR-10 | ResNet35 | $94.6_{\pm0.34}$ | $94.8_{\pm0.35}$ | $94.9_{\pm0.39}$ | $95.2_{\pm0.40}$ | $95.3_{\pm0.42}$ | $\mathbf{95.6}_{\pm0.36}$ | $95.2_{\pm0.45}$ | $94.9_{\pm0.41}$ |
| CIFAR-10 | VGG16 | $93.0_{\pm0.40}$ | $93.2_{\pm0.39}$ | $93.2_{\pm0.43}$ | $93.4_{\pm0.45}$ | $93.6_{\pm0.52}$ | $93.8_{\pm0.40}$ | $\mathbf{94.1}_{\pm0.45}$ | $93.7_{\pm0.45}$ |
| CIFAR-10 | MobileNetV1 | $92.9_{\pm0.43}$ | $93.0_{\pm0.47}$ | $93.6_{\pm0.49}$ | $93.8_{\pm0.51}$ | $93.9_{\pm0.51}$ | $\mathbf{94.1}_{\pm0.46}$ | $93.8_{\pm0.53}$ | $93.9_{\pm0.46}$ |
| CIFAR-100 | ResNet18 | $74.3_{\pm0.21}$ | $73.8_{\pm0.20}$ | $73.6_{\pm0.29}$ | $73.4_{\pm0.32}$ | $73.1_{\pm0.33}$ | $\mathbf{74.6}_{\pm0.27}$ | $73.4_{\pm0.28}$ | $73.7_{\pm0.31}$ |
| CIFAR-100 | ResNet35 | $74.6_{\pm0.19}$ | $74.0_{\pm0.23}$ | $73.8_{\pm0.30}$ | $73.7_{\pm0.33}$ | $73.4_{\pm0.31}$ | $\mathbf{74.7}_{\pm0.27}$ | $73.9_{\pm0.29}$ | $74.1_{\pm0.30}$ |
| CIFAR-100 | VGG16 | $72.4_{\pm0.33}$ | $72.3_{\pm0.36}$ | $71.9_{\pm0.34}$ | $71.8_{\pm0.39}$ | $71.3_{\pm0.40}$ | $\mathbf{72.6}_{\pm0.31}$ | $71.8_{\pm0.39}$ | $72.0_{\pm0.37}$ |
| CIFAR-100 | MobileNetV1 | $73.9_{\pm0.42}$ | $\mathbf{74.1}_{\pm0.46}$ | $73.7_{\pm0.50}$ | $73.3_{\pm0.49}$ | $72.7_{\pm0.42}$ | $73.8_{\pm0.43}$ | $73.1_{\pm0.44}$ | $73.9_{\pm0.46}$ |
| Tiny-ImageNet | ResNet34 | $54.2_{\pm0.18}$ | $53.9_{\pm0.20}$ | $53.5_{\pm0.21}$ | $53.6_{\pm0.21}$ | $53.6_{\pm0.25}$ | $\mathbf{54.5}_{\pm0.32}$ | $54.3_{\pm0.29}$ | $52.9_{\pm0.42}$ |

Table 3: Model calibration of different gradient decay and post-processing calibration. The best results are in bold. Results are averaged over five runs with different seeds. ($bins = 10$)

| Dataset | Model | Metric | Gradient decay factor $\beta$ | | | | Vector Scaling | Temp. Scaling |
|---|---|---|---|---|---|---|---|---|
| | | | 20 | 10 | 1 | 0.1 | | |
| CIFAR-100 | ResNet18 | ECE | $\mathbf{0.019}_{\pm0.003}$ | $0.048_{\pm0.008}$ | $0.111_{\pm0.011}$ | $0.161_{\pm0.021}$ | $0.039_{\pm0.006}$ | $0.026_{\pm0.005}$ |
| | | MCE | $\mathbf{0.063}_{\pm0.011}$ | $0.139_{\pm0.025}$ | $0.306_{\pm0.051}$ | $0.423_{\pm0.076}$ | $0.135_{\pm0.036}$ | $0.064_{\pm0.021}$ |
| CIFAR-100 | ResNet34 | ECE | $\mathbf{0.026}_{\pm0.004}$ | $0.055_{\pm0.008}$ | $0.131_{\pm0.019}$ | $0.182_{\pm0.022}$ | $0.042_{\pm0.006}$ | $0.038_{\pm0.005}$ |
| | | MCE | $0.087_{\pm0.011}$ | $0.162_{\pm0.031}$ | $0.233_{\pm0.068}$ | $0.332_{\pm0.091}$ | $0.131_{\pm0.032}$ | $\mathbf{0.059}_{\pm0.011}$ |
| CIFAR-100 | VGG16 | ECE | $0.122_{\pm0.009}$ | $0.163_{\pm0.011}$ | $0.207_{\pm0.031}$ | $0.226_{\pm0.033}$ | $0.030_{\pm0.008}$ | $\mathbf{0.022}_{\pm0.005}$ |
| | | MCE | $0.317_{\pm0.021}$ | $0.378_{\pm0.051}$ | $0.499_{\pm0.088}$ | $0.556_{\pm0.093}$ | $0.523_{\pm0.109}$ | $\mathbf{0.041}_{\pm0.011}$ |
| CIFAR-10 | ResNet18 | ECE | $0.021_{\pm0.004}$ | $0.025_{\pm0.006}$ | $0.036_{\pm0.010}$ | $0.042_{\pm0.011}$ | $\mathbf{0.011}_{\pm0.003}$ | $0.015_{\pm0.003}$ |
| | | MCE | $0.591_{\pm0.153}$ | $0.268_{\pm0.095}$ | $0.295_{\pm0.068}$ | $0.355_{\pm0.111}$ | $\mathbf{0.051}_{\pm0.012}$ | $0.089_{\pm0.009}$ |
| Tiny-ImageNet | ResNet34 | ECE | $\mathbf{0.014}_{\pm0.002}$ | $0.036_{\pm0.005}$ | $0.089_{\pm0.011}$ | $0.226_{\pm0.056}$ | $0.017_{\pm0.006}$ | $0.019_{\pm0.004}$ |
| | | MCE | $\mathbf{0.035}_{\pm0.005}$ | $0.069_{\pm0.009}$ | $0.166_{\pm0.021}$ | $0.382_{\pm0.079}$ | $0.036_{\pm0.010}$ | $0.063_{\pm0.013}$ |
| Tiny-ImageNet | ResNet50 | ECE | $0.041_{\pm0.007}$ | $\mathbf{0.013}_{\pm0.002}$ | $0.104_{\pm0.021}$ | $0.188_{\pm0.032}$ | $0.023_{\pm0.004}$ | $0.027_{\pm0.003}$ |
| | | MCE | $0.082_{\pm0.011}$ | $0.044_{\pm0.009}$ | $0.149_{\pm0.020}$ | $0.377_{\pm0.045}$ | $\mathbf{0.039}_{\pm0.011}$ | $0.067_{\pm0.016}$ |

deviation between the original cross-entropy loss and objective functions with different $\beta$. Significantly, the result of this experiment is enough to empirically show that minor gradient decay with small $L$-constraint can achieve a faster convergence rate.

The top-1 accuracy of several deep models are given in Tab. 2. It can be concluded that top-1 accuracy of these models benefit from the minor gradient decay in SVHN and CIFAR-10. However, with over-small gradient decay, some challenging samples may remain low confidence $\hat{p}$. So as shown in Tab. 2, over-small $\beta$ cannot indicate a better performance.

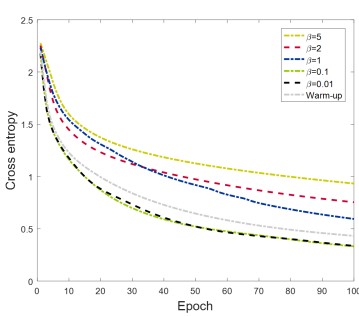

On the other hand, a large $\beta$ induces a large gradient decay and a large $L$-constraint of Softmax, which means that the probability output of easy samples and challenging samples keep relatively consistent, as discussed in Fig. 3. Besides, a small gradient magnitude could lead to insufficient training, as demonstrated in Tab. 2. This could elucidate the subpar outcomes associated with large gradient decay coefficients in CIFAR-10 and SVHN datasets.

Figure 5: CE with different gradient decay.

However, confidence is not always a good thing. The design of the curriculum, specifically in evaluating the difficulty of individual samples, plays a pivotal role in curriculum learning idea. Apart from the manual design, most methods can only be done by the posterior probability. Based the previous analysis in section 3.2, over-small gradient decay rate may impair the performance since some challenging samples containing important information remain low confidence $\hat{p}$. As shown in Tab. 2, these models on CIFAR-100 all prefer the large $\beta$. From another perspective, the mentioned model has no high probability confidence and the training material is not good enough. Learning every sample more equally is a better choice since the model cannot confidently determine which the simple sample is or which a more informative sample is by posterior

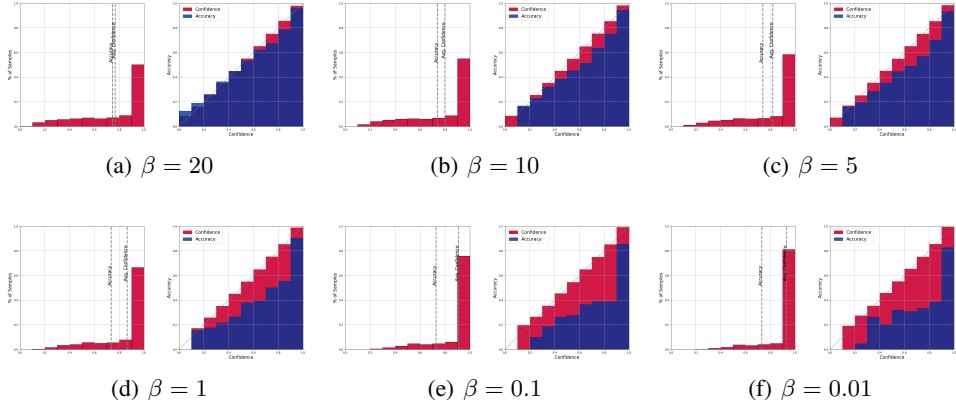

Figure 6: **Confidence and reliability diagrams with ResNet18 on CIFAR-100.** ($bins = 10$) In each subplot, the left plot illustrates the sample distribution in individual bins, while the right plot displays the average confidence and accuracy in each bin. Ideally, calibration aims for consistency between accuracy and average confidence in each bin. It indicates that a smaller gradient decay rate Beta is associated with more pronounced miscalibration of the model, while a larger gradient decay rate mitigates this issue.

probability. The large gradient decay (Liu et al., 2016; 2017; Wang et al., 2018) is a relative concept in different tasks.

## 4.2 MODEL CALIBRATION

To substantiate this conclusion on the model calibration, Fig. 6 displays the confidence histograms and reliability diagrams for different gradient descent factors on CIFAR-100 while Tab. 3 provides the ECE and MCE in different datasets, which are crucial metrics for assessing model calibration (Guo et al., 2017). The results reveal that small decay coefficients, corresponding to larger margin penalties in Softmax, result in overconfidence, rendering the probabilistic output less reliable. Conversely, a significant probability-dependent gradient decay mitigates model overconfidence.

Experimental results consistently demonstrate that as the gradient decay rate decreases with rising probabilities, the average confidence also rises. This phenomenon can be attributed to the small gradient decay rate enforcing a strict curriculum learning sequence. The adjustment of probability-dependent gradient decay would significantly improve the confidence distribution in model training, surpassing some post-calibration methods even with rough tuning on CIFAR-100 and Tiny-ImageNet. We attribute the poor calibration to the small gradient decay rate, and this conclusion is compelling. More experimental results supporting this conclusion can be found in Appendix A.

## 5 CONCLUSION

This paper introduces the gradient decay hyperparameter $\beta$ to analyze the effect of the large margin defined in decision space from a dynamical training process. The large margin of Softmax induces the small gradient decay as the sample probability rises. The easy positive samples can be learned sufficiently up to high probability and the model tends to be more confident toward these samples. Training displays distinguishability over different samples in training, i.e., the samples are optimized under the stricter curriculum sequence. Under the limited network capacity and over-large margin, reducing the intra-class distance of the partial easy positive samples will sacrifice the inter-class distance of hard negative samples. Empirical evidence demonstrates that small probabilistic gradient decays exacerbate the miscalibration of over-parameterized models. Conversely, increasing the gradient decay coefficient emerges as an effective strategy for alleviating issues related to overconfidence. Besides, the Softmax with smaller gradient decay has a smoother local $L$-constraint, so the large margin Softmax can obtain a faster convergence rate. Thus, we propose a warm-up training strategy to smoother $L$-constraint in early training and avoid over-confidence in final model.

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

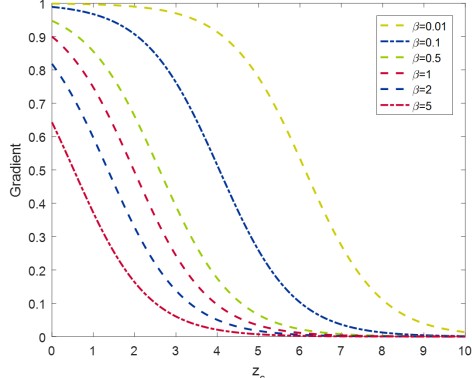

Figure 7: The gradient magnitude as the sample output increases with different hyperparameter values under the assumption: 1) Initialization $z_j = 0, j = 1, \ldots, m$ and $m = 10$  2) Other class outputs are equal, i.e., $\frac{\partial J}{\partial z_j} = -\frac{1}{m-1}\frac{\partial J}{\partial z_c}, j = 1, \ldots, m, j \neq c$.

## A  APPENDIX

### A.1  IMPLEMENTATION DETAILS IN EXPERIMENTS

In this experiment, we give some other experimental results of a variety of model architectures, such as FCNN, ResNet18, ResNet35, ResNet50 He et al. (2016), VGG16 Simonyan & Zisserman (2014) and MobileNetV1 Howard et al. (2017), trained on MNIST, CIFAR-10/100, SVHN and Tiny-ImageNet. Based on empirical analysis, we briefly show how different $\beta$ dynamical affect the performance of the models on different datasets.

The shallow three-layer architecture of FCNN with 50, 20 and 10 nodes is implemented on MNSIT. The learning rate is $10^{-3}$ and weight decay is $10^{-4}$. The momentum is set to 0.9. The batch is set to 100 within a total of 100 epochs. $\beta_{initial}$ and $\beta_{end}$ in the warm-up strategy are set to 0.01 and 0.1, respectively, where over-small $\beta = 0.001$ provides faster convergence in early phase.

The deep models are used for training and predicting in CIFAR-10/100, SVHN and Tiny-ImageNet. SGD is set as the optimizer. The learning rate is $10^{-2}$ and weight decay is $10^{-4}$. The momentum is set to 0.9. The batch is set to 100. CIFAR-10/100 is traversed over 200 epochs, while SVHN is traversed over 100 epochs. At 50% and 75% epochs, the learning rate decreases to 10 percent. At the same time, we set gradient clip with $norm = 3$. For the Tiny-ImageNet, the learning rate was set to 0.05, the batch size to 256, the momentum to 0.9, and the weight clipping to Norm=3. 200 epochs were performed, with the learning rate decreasing to 10% at 40% and 80% epochs, respectively. $\beta_{initial}$ and $\beta_{end}$ in the warm-up strategy are set to 0.1 and 1, respectively. In real-world training, we find that over-small $\beta$ may cause the overflow of the Softmax function. So $e^z$ is replaced by $e^{z-u}$ in (19) and $u$ is set to 70 to prevent exp function from being too large in experiments. All results are averaged over five runs with different seeds using 'torch.manual_seed(5)' to 'torch.manual_seed(9)'.

$$J = -\log \frac{e^{z_c - u}}{\sum_{j \neq c} e^{z_j - u} + \beta e^{z_c - u}} \tag{19}$$

### A.2  ADDITIONAL RESULTS

We can observe that there is a noticeable accuracy oscillation in Fig. 8. In experiments, we find that it is a common phenomenon in different datasets when $\beta$ is large but this kind of large consolidation does not happen when momentum is small. We attribute this phenomenon to instability resulting from rapid gradient decay in certain samples. To provide a clearer description of this issue, further systematic exposition and experimentation are needed.

In the experiments detailed in Table 4, we investigated the impact of larger gradient decay coefficients on model calibration and generalization in the CIFAR-100. Our findings indicate that

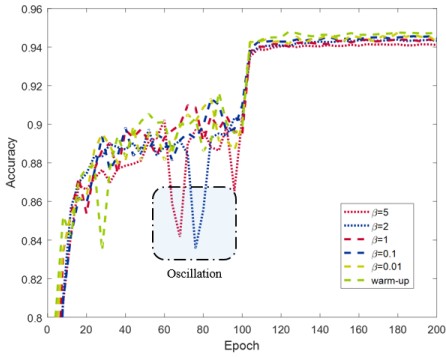

Figure 8: Top-1 accuracy curve on CIFAR-100 with different gradient decay hyperparameters {5, 2, 1, 0.1, 0.01} and warm-up strategy.

Table 4: Top-1 Acc (%), ECE and MCE ($bins = 10$) on CIFAR-100 with ResNet17 and ResNet34

| Dataset | Model | Metric | Gradient decay factor $\beta$ | | | | | |
|---------|-------|--------|------|------|------|------|------|------|
| | | | 50 | 20 | 10 | 5 | 1 | 0.1 |
| CIFAR-100 | ResNet18 | Top-1 Acc | **75.0** | 74.5 | 74.2 | 74.1 | 73.5 | 73.2 |
| | | ECE | 0.027 | **0.021** | 0.059 | 0.082 | 0.130 | 0.181 |
| | | MCE | 0.141 | **0.071** | 0.135 | 0.182 | 0.298 | 0.464 |
| CIFAR-100 | ResNet34 | Top-1 Acc | **75.6** | 75.2 | 74.8 | 74.5 | 74.0 | 73.8 |
| | | ECE | **0.024** | 0.030 | 0.065 | 0.085 | 0.125 | 0.170 |
| | | MCE | **0.056** | 0.091 | 0.157 | 0.172 | 0.273 | 0.388 |

Table 5: The performance of ResNet34 on Tiny-ImageNet with different gradient decay.

| Metric | Gradient decay factor $\beta$ | | | | | |
|--------|------|------|------|------|------|------|
| | 20 | 10 | 5 | 1 | 0.1 | 0.01 |
| Top-1 Acc (%) | 52.8 | 53.2 | **53.8** | 53.4 | 53.7 | 53.6 |
| Top-5 Acc (%) | **75.8** | 75.5 | 75.1 | 75.0 | 74.4 | 73.2 |
| Training Acc (%) | 88.6 | 88.5 | 89.7 | 90.3 | **90.9** | 90.5 |
| ECE ($bins = 10$) | **0.015** | 0.034 | 0.076 | 0.087 | 0.224 | 0.274 |
| MCE ($bins = 10$) | **0.036** | 0.065 | 0.151 | 0.176 | 0.406 | 0.518 |

Table 6: The performance of ResNet50 on Tiny-ImageNet with different gradient decay.

| Metric | Gradient decay factor $\beta$ | | | | | |
|--------|------|------|------|------|------|------|
| | 20 | 10 | 5 | 1 | 0.1 | 0.01 |
| Top-1 Acc (%) | 55.9 | 56.0 | **56.4** | 56.3 | 56.4 | 56.2 |
| Top-5 Acc (%) | 77.7 | **78.0** | 77.3 | 76.6 | 76.0 | 74.9 |
| Training Acc (%) | 86.4 | 88.8 | 90.3 | 91.9 | **92.3** | 91.8 |
| ECE ($bins = 10$) | 0.045 | **0.014** | 0.046 | 0.114 | 0.203 | 0.249 |
| MCE ($bins = 10$) | 0.084 | **0.044** | 0.082 | 0.151 | 0.388 | 0.476 |

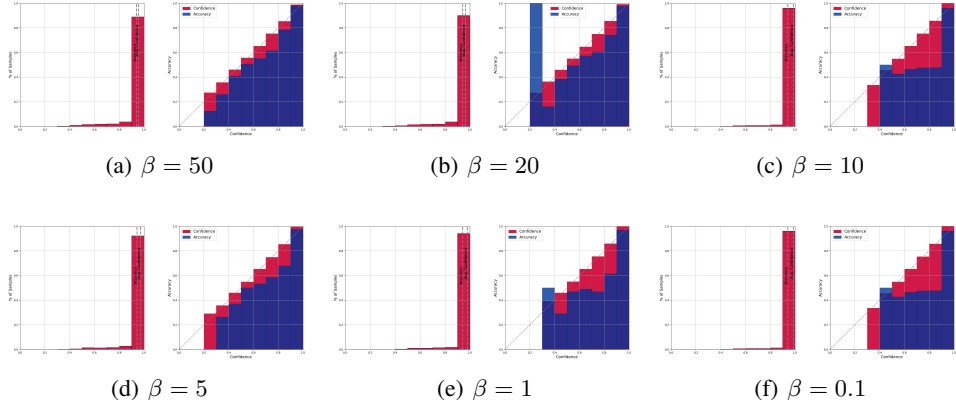

Figure 9: Confidence histograms and reliability diagrams for gradient decay with ResNet18 on CIFAR-10. ($bins = 10$)

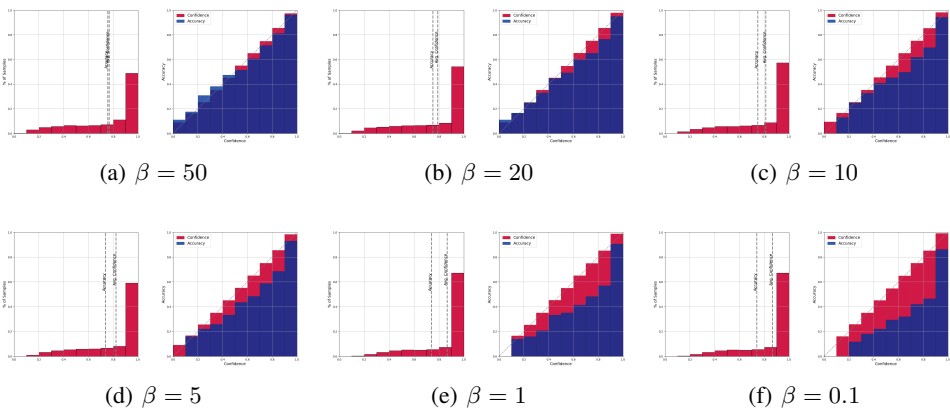

Figure 10: Confidence histograms and reliability diagrams for gradient decay with ResNet34 on CIFAR-100. ($bins = 10$)

as the gradient decay coefficients increase, the overall confidence level decreases, resulting in the model exhibiting underconfidence. Remarkably, despite the model under-confidence, as indicated by the ECE calibration metric, its accuracy continues to improve significantly. This suggests that for CIFAR-100 with ResNet18 and ResNet34 architectures, training consistency between samples is favored.

The results in Fig. 12 and Table 5 demonstrate that smaller gradient decay weights lead to faster convergence speeds, resulting in higher training accuracy. This can be attributed to the imposition of smaller local L constraints by the low gradient decay rate, which in turn increases gradient magnitude. On the other hand, higher gradient decay rates promote optimization consistency and yield better values for metrics such as ECE, MCE, and Top-5 Accuracy. However, these higher gradient decay rates, characterized by smaller gradient magnitudes, do not necessarily lead to improved Top-1 accuracy. Figs. 9-12 consistently demonstrate that larger gradient decay rates lead to improved model confidence.

## A.3 DISCUSSION

The penalization of larger margin Softmax leads to a deceleration in the gradient decay rate, which exerts influence on amples throughout the dynamics training process. Smaller gradient decay enforce a stricter curriculum learning sequence, thereby augmenting the confidence of easier samples.

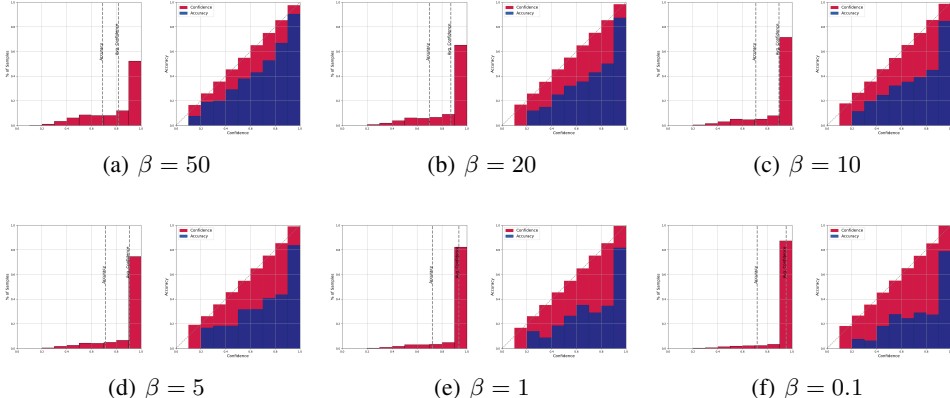

(a) $\beta = 50$      (b) $\beta = 20$      (c) $\beta = 10$

(d) $\beta = 5$      (e) $\beta = 1$      (f) $\beta = 0.1$

Figure 11: Confidence histograms and reliability diagrams for gradient decay with VGG16 on CIFAR-100. ($bins = 10$)

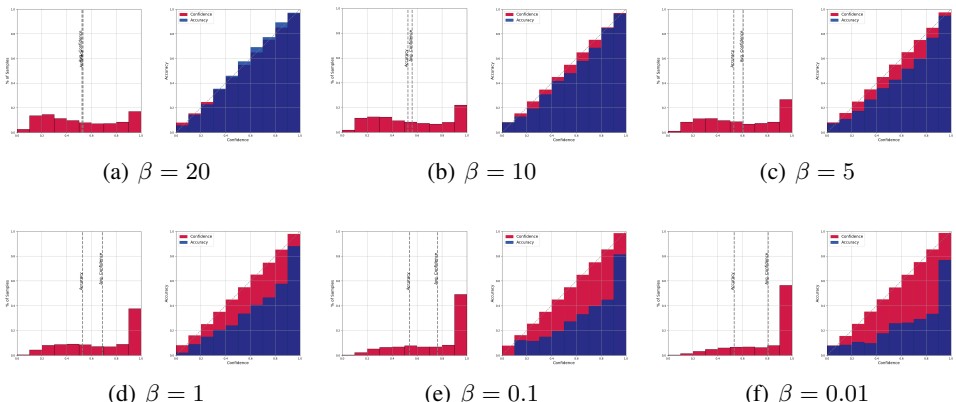

(a) $\beta = 20$      (b) $\beta = 10$      (c) $\beta = 5$

(d) $\beta = 1$      (e) $\beta = 0.1$      (f) $\beta = 0.01$

Figure 12: Confidence histograms and reliability diagrams for gradient decay with ResNet34 on Tiny-ImageNet. ($bins = 10$)

Conversely, challenging samples are more susceptible to being overlooked due to the learning sequence, ultimately culminating in poor model calibration.

In all experiments, despite the consistent application of an identical learning rate to control the scalar method and draw scientifically grounded conclusions, we inadvertently neglected to consider the impact of varying gradient magnitudes in the learning process. As illustrated in Table 2, a high gradient decay rate may yield insufficient learning outcomes for certain samples, resulting in undesirable performance. In essence, the gradient decay rate not only impacts the learning sequence among individual samples but also influences the gradient magnitude concerning the overall sample learning process. To attain more scientifically robust conclusions, it becomes imperative to employ more meticulously designed experiments and generate corroborating results.

As such, one potential direction for future research is to employ more rigorously designed experiments to probe into the dynamic properties of gradient decay coefficients during the optimization process. Expanding on the previous discussion, there is a promising avenue for technical improvement related to designing decay coefficients or incorporating adaptive mechanisms during training. Besides, a high gradient decay rate can result in an overall gradient that becomes excessively small, thereby hindering the training process. Developing methods specifically designed to adjust learning rates based on gradient decay coefficients holds the potential to significantly enhance the efficiency of model training procedures. It ultimately enables the modification of the learning rate scheduler to

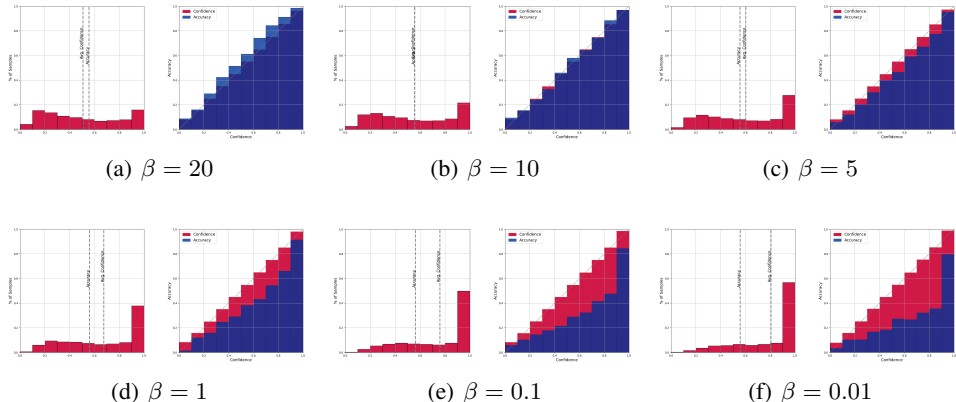

(a) $\beta = 20$      (b) $\beta = 10$      (c) $\beta = 5$

(d) $\beta = 1$      (e) $\beta = 0.1$      (f) $\beta = 0.01$

Figure 13: Confidence histograms and reliability diagrams for gradient decay with ResNet50 on Tiny-ImageNet. ($bins = 10$)

achieve effective model calibration with the large gradient decay and to prevent under-training due to small gradients.

Furthermore, we posit that the distinct patterns exhibited by varying gradient decay coefficients across different experiments are closely linked to the model's capacity and the dataset's level of complexity. These conclusions warrant a more systematic and rigorous discussion.

