# OpenReview forum: "Probability-dependent gradient decay in large margin softmax"
_ICLR.cc/2024/Conference — Submitted to ICLR 2024_

### Official Review · Reviewer_kFkh · 2023-10-31

**Soundness:** 2 fair
**Presentation:** 1 poor
**Contribution:** 2 fair
**Rating:** 5
**Confidence:** 3

**Summary:**

This paper studies the impact of adding a $\beta$ term in the softmax equation. Let $m$ be the number of classes, they propose:
$$p = \frac{e^x}{\sum_i e^{x_i} + \beta e^x}, \quad \text{with } x \in R^{m}, \text{and } p \in R^{m}$$
When used alongside the cross-entropy loss, the formulation of the loss on a training sample $(x,y)$ becomes:
$$\ell(x,y) = -\log(p_y) = - \log(\frac{e^{x_y}}{\sum_i e^{x_i} + \beta e^{x_y}}) $$
They show how $\beta$ enforces a soft margin and modulates the gradient magnitude depending on the probability $p_y$. More precisely, they show how small $\beta$s increase the gradient magnitude for larger probabilities---promoting a soft margin---, while larger $\beta$s reduce the gradient magnitude for larger probabilities. They theoretically derive this observation and validate it empirically on $4$ vision datasets. Moreover, they draw a parallel with curriculum learning and calibration.

**Strengths:**

I find this work well motivated: the softmax function is ubiquitous in modern machine learning and studying its various caveats is important.
The connection with calibration is interesting, and the results in figure 6 are very promising. Especially, the calibration improves as $\beta$ increases, which allows the model to be less influenced by current samples having a $p_y$ close to $1$.

**Weaknesses:**

I found two main weaknesses in this work. The first one consists of the overall lack of clarity. I find the paper hard to read. Here are some parts I found confusing:
- "MSE takes into account more complex optimization scenarios": What do you mean by that?
- "Hard mining strategy": you could briefly introduce what this is.
- in section 2, you talk about $J_j$ before introducing it
- In figure 3: there are no legends for the top row, and the caption does not help to clarify the different curves being shown, the main text is also unclear about those i.e. what are "post-training samples", are those test samples?
- In figure 3 still, it should be mentioned in the legend or in the caption that the different groups correspond to samples of varying difficulty
- "If we make excessive demands on the margin, some post-training samples cannot get any chance and will be "sacrificed" according to the soft curriculum learning strategy [...]", what do you mean?
- "So it is convinced that the general softmax [...]"
- "The beta smaller is, the gradient smoother is"
- "Warm-up strategy achieves even better results."
- "Curriculum design that divides samples is crucial to curriculum learning idea."
- "Besides, based the previous analysis [...]"
- In figure 6: the y axis mentions accuracy and but also shows confidence, this is confusing and could be clarified in the caption.

The second weakness is the lack of rigor in the experiments:
- In figure 4: which model is being used? How many parameters? How were the hyperparameters tuned? Are those averaged over multiple seeds, if yes can we see the standard deviation?
- In table 2: The different values are quite close, and it is difficult to evaluate the robustness of the improvement without standard deviation. It should be possible to run the same experiment with different seeds for some of the smaller datasets.
- In table 3: same as above, I would love to see standard deviations
- For all the experiments, which experimental protocol was followed: which architecture, tuning, seeds, optimizer, ... I couldn't find those in the appendix either

**Questions:**

See above.

---

> ### Author Response · Authors · 2023-11-17
> **Response to Reviewer kFkh**
>
> Thank you to the reviewer for taking the time to evaluate our paper and providing constructive the suggestion.
>
> Following the reviewer's guidance, we have recognized the poor presentation of our paper. Considering the readability of the paper, we have modified the presentation in section1 and section2 to improve presentation of the paper. In the original version, certain statements were confusing. We have made corresponding revisions to enhance clarity. Additionally, in response to the reviewer's advice, we have supplemented the standard deviation of the results in our experiments.
>
> We have provided detailed point-to-point responses to the reviewer's inquiries as followings.
> * Q1: "MSE takes into account more complex optimization scenarios": What do you mean by that?
> * A1: We have recognized that the meaning of this sentence was ambiguous. In the current version of the paper, this statement has been replaced with an introduction to model calibration, aiming to better elucidate the content presented in the paper.
>
> ”However, the modern models often demonstrate inadequate confidence calibration in probability distribution through Softmax mapping. Specifically, these probability outputs display unwarranted over-confidence (Guo et al., 2017). Furthermore, researchers have identified that achieving high accuracy in classifiers and calibrating the model confidence are distinct objectives (Wenger et al., 2020). This scenario emphasizes the necessity to study the calibration of model output uncertainties in optimization.”
>
> * Q2: "Hard mining strategy": you could briefly introduce what this is.
> * A2: Thank you for the suggestion. We have briefly introduced the hard mining strategy in the third paragraph of Section 1.
>
> * Q3: In section 2, you talk about J_i before introducing it
> * A3: We have noticed the poorly articulated expression in Section 2. In the current version, we have polished Eq. (1)-(6). Furthermore, we have added explanations regarding J in Eq. (1).
>
> * Q4: In figure 3: there are no legends for the top row, and the caption does not help to clarify the different curves being shown, the main text is also unclear about those i.e. what are "post-training samples", are those test samples?
> * A4: The term "post-training samples" refers to samples with low confidence in the early stages of training. We have revised the confusing expression to use "challenging samples." Additionally, we have updated the captions for each figure to better describe the content of the figures.
>
> * Q5: "If we make excessive demands on the margin, some post-training samples cannot get any chance and will be "sacrificed" according to the soft curriculum learning strategy [...]", what do you mean?
> * A5: This sentence indicates that certain samples consistently exhibit low confidence when the margin is set too large. We have replaced this confusing statement with the following illustration.
>
> “If we make excessive demands on the margin, some challenging samples always remain low confidence $\hat p$ according to the stringent curriculum learning sequence.”
>
> * Q6: "So it is convinced that the general softmax [...]"
> * A6: The meaning of this sentence is that Softmax could be characterized by two hyperparameters, $\tau$ and $\beta$. In the current version, we have replaced some inappropriate terms, as illustrated by the following sentence:
>
> “So it is convinced that Softmax cross-entropy could be characterized with these two hyperparameters $\tau $ and $\beta$.”
>
> * Q7: "The beta smaller is, the gradient smoother is"
> * A7: Regarding the function $J\left( {{z_c}} \right)$ presented in equations (9)-(12), we determine that $\max {\left| {\nabla _{{z_c}}^2J\left( {{z_c}} \right)} \right|_2}$ equals a constant when ${p_c} = \frac{1}{{1 + \beta }}$. By considering the local range ${p_c} \in \left[ {0,0.5} \right]$, we can derive a local $L$-constraint for ${\left| {\nabla _{{z_c}}^2J\left( {{z_c}} \right)} \right|_2}$, as illustrated in equation (18). The original sentence was inaccurately expressed. We have replaced this confusing sentence with the following statement.
>
> “The $\beta$ smaller is, L-constrain in the early phase of optimization smaller is.”
>
> * Q8: "Warm-up strategy achieves even better results."
> * A8: As depicted in Figure 4, the warm-up strategy yields higher accuracy compared to a fixed gradient decay coefficient. We have replaced the original confusing sentence with the following statement:
>
> "As a result, the warm-up strategy achieves higher accuracy compared to a fixed gradient decay coefficient."

---

> ### Author Response · Authors · 2023-11-17
> **Response to Reviewer kFkh**
>
> * Q9: "Curriculum design that divides samples is crucial to curriculum learning idea."
> * A9: Thank you for the reviewer's thorough examination. The original sentence aimed to convey that the design of the curriculum necessitates distinguishing between easy and challenging samples, which is crucial for the entire curriculum learning. We acknowledge the poor expression in the original sentence and have replaced it with the following clarification:
>
> “The design of the curriculum, specifically in evaluating the difficulty of individual samples, plays a pivotal role in curriculum learning idea.”
>
> * Q10: "Besides, based the previous analysis [...]"
> * A10: The original sentence aimed to convey that these challenging samples may exhibit low confidence when the gradient decay rate is excessively small. We recognize the inaccuracies in the original wording, and in the current version, this confusing sentence has been replaced with the following clarification.
>
> “Based the previous analysis in section 4.2, over-small gradient decay rate may impair the performance since some challenging samples containing important information remain low confidence $\hat p$.”
>
> * Q11: In figure 6: the y axis mentions accuracy and but also shows confidence, this is confusing and could be clarified in the caption.
> * A11:  Thank you for your suggestions. We have revised the caption of figure 6 to provide more detailed descriptions of their content. Additionally, to better introduce the concept of model calibration, we have incorporated relevant descriptions into section 1 and section 2. The modified caption is presented below:
>
> “Confidence and reliability diagrams with ResNet18 on CIFAR-100. ($bins=10$) In each subplot, the left plot illustrates the sample distribution in individual bins, while the right plot displays the average confidence and accuracy in each bin. Ideally, calibration aims for consistency between accuracy and average confidence in each bin. It indicates that a smaller gradient decay rate $\beta$ is associated with more pronounced miscalibration of the model, while a larger gradient decay rate mitigates this issue.”
>
> * Q12: In figure 4: which model is being used? How many parameters? How were the hyperparameters tuned? Are those averaged over multiple seeds, if yes can we see the standard deviation?
> * Q13: In table 2: The different values are quite close, and it is difficult to evaluate the robustness of the improvement without standard deviation. It should be possible to run the same experiment with different seeds for some of the smaller datasets.
> * Q14: In table 3: same as above, I would love to see standard deviations.
> * Q15: For all the experiments, which experimental protocol was followed: which architecture, tuning, seeds, optimizer, ... I couldn't find those in the appendix either.
> * A12 to Q12-15: We sincerely appreciate the constructive suggestions from the reviewer. The model employed in Fig. 4 is a three-layer fully connected network. Detailed parameter settings and all experimental details, including architecture, tuning, seeds, and optimizer, are provided in Appendix A.1. To offer more general conclusions, all hyperparameters are set without cumbersome tuning. In addition, all experimental results are averages of five different random seeds and the specific random seeds are given in Appendix A.1. The random seed configurations were consistent across different methods. Furthermore, following reviewer's recommendation, we have included standard deviation in the results of repeated experiments in Tables 2 and 3.

---

> ### Author Response · Authors · 2023-11-17
> **Response to Reviewer kFkh**
>
> Dear Reviewer kFkh,
>
> Thank you for your valuable feedback on our submission. We have read your comments carefully and have addressed them in our rebuttal. We would be grateful if you could acknowledge if our responses have addressed your comments. Thank you again for your time and consideration.
>
> Best regards,
>
> Authors of paper 1676.

---

### Official Review · Reviewer_KFsP · 2023-11-01

**Soundness:** 3 good
**Presentation:** 3 good
**Contribution:** 2 fair
**Rating:** 5
**Confidence:** 3

**Summary:**

The paper studies the influence of introducing a margin parameter $\beta$ in the softmax operator for classification problems. The idea is to relate the margin parameter $\beta$ to the decay rate of the gradients so that the classification confidence can be manipulated. The margin parameter also gives rise to some improved performance in image data. I would view the margin parameter $\beta$ as the major contribution of the paper, as existing study often focuses on the temperature parameter.

**Strengths:**

The paper provides a very detailed guide to understand the influence of the margin parameter $\beta$ on the decay rate of the gradient. The study looks comprehensive and correct, which leads to the successful empirical verification.

Most of the paper is well organized, although some part needs additional care.

**Weaknesses:**

Section 2 needs a revision. See more in questions section.

As far as I can tell, the classification error improvement is a bit marginal. The baseline accuracy should correspond to $\beta = 1$ and the highlighted best obtained errors may not have significant improvement. Although this evaluation might be objective, but this concern can be partially addressed by providing a standard deviation computed in multiple runs, so that the statistical significance can be verified.

Given the concerns above, I am giving a negative rating. However, I am willing to discuss with the authors on the significance of the proposed method and potentially raise the score.

**Questions:**

The grammar around Equations (1) -- (6) should be polished.

In Equation (4), (5) and (6), is there a bracket around $z_i - z_c$?

What is hidden in the approximate equality in Equation (5)?

Figure 3 has a vague description: "confidence of some samples during training". The font in the figures is small.

---

> ### Author Response · Authors · 2023-11-17
> **Response to Reviewer KFsP**
>
> We appreciate the reviewer for taking the time to evaluate our paper and providing constructive the suggestion.
>
> * We appreciate the suggestions and questions raised by the reviewer under "Weaknesses" and "Questions." Following the reviewer's guidance, we recognized deficiencies in the presentation in Section 2. We have revised the presentation in Section 2 to better elucidate "Preliminaries". Eq. (1) to Eq. (6) have been polished for a clearer presentation.
>
> * Eq. (4), (5), and (6) in the original manuscript have been refined and now appear as Eq. (3) in the current version. This presentation demonstrates an approximation between Softmax cross-entropy and the max function, contributing to a better understanding of the character of the two hyperparameters.
>
> * In addition, we have adjusted the font size of figure 3 and other figures to more clearly convey the meaning of the visuals. Besides, we have revised the caption of figures to provide more detailed descriptions of their content. In experiments, we also present the average results under five different random seeds and the corresponding standard deviations in Tables 2 and 3 to support our conclusions and improve the rigor of the experiment.

---

> ### Author Response · Authors · 2023-11-17
> **Response to Reviewer KFsP**
>
> We appreciate the reviewer's willing in discussing the significance of our paper. In response, we would like to provide additional explanations about the significance of our work.
>
> As mentioned by the reviewer in “Weaknesses”, the warm-up scheme shows an improvement over the previously proposed adjustment for Softmax, but the improvement is not significant.
> Following review’s guidance and concerns, we repeated the results more times and presented the mean and standard deviation of the results in Tab.2 and Tab.3 to support our conclusions. Considering that these two approaches address the problem from different perspectives, where A-Softmax statically considers the margin in the output space, while the warm-up scheme dynamically adjusts gradient decay coefficients during the optimization process, we believe that the presentation of such results can provide some new insights.
>
> On the other hand, our paper also focuses on the uncertainty estimation of the classification results, i.e., whether the model probability outputs can reflect the accuracy prediction in classification. Model calibration is important in practical applications. Perfect calibration of neural network can be realized when the confidence score reflects the real probability that the classification is classified correctly. Formally, the perfectly calibrated network satisfied ${\rm P}\left( {\hat y = y|\hat p = p} \right) = p$ for all $p \in \left[ {0,1} \right]$. However, probability outputs of modern model display unwarranted over-confidence. Our work reveals that probability-dependent decay rate is a factor contributing to the over-confidence of modern models in classification. The optimization with a small gradient decay rate presents a more stringent curriculum learning for samples, meaning that challenging samples only gain confidence after easy samples have been sufficiently trained to a certain level of confidence. This leads to different samples having greater distinctiveness in the optimization. Correspondingly, our presented analysis, detailed in Section 3, along with more experimental results shown in Fig. 6, Table 3, and Appendix, demonstrates that a substantial increase in the model-calibrated criterion ECE and MCE can be achieved when employing large probability-dependent gradient decay in modern model. To better present the work in model calibration, we have modified presentation in section 1 and section 2 to emphasize this point.
>
> In addition to methodological innovations, we think that our work can provide some new insights and understanding, especially concerning concepts such as model calibration, large-margin Softmax, probability-related gradient decay, curriculum learning, and related topics in deep learning optimization.

---

> > ### Author Response · Authors · 2023-11-17
> > **Response to Reviewer KFsP**
> >
> > Dear Reviewer KFsP,
> >
> > Thank you for your valuable feedback on our submission. We have read your comments carefully and have addressed them in our rebuttal. We would be grateful if you could acknowledge if our responses have addressed your comments. Thank you again for your time and consideration.
> >
> > Best regards,
> >
> > Authors of paper 1676.

---

> > ### Comment · Reviewer_KFsP · 2023-11-23
> >
> > Dear Authors,
> >
> > Sorry for a late reply. I really appreciate your detailed response and additional experimental results, as well as the revisions in the paper. With the standard deviation in Table 1 and 2, I can see some improvements are indeed statistically significant and the value of the method is better supported. Therefore, I raised the score to a 5 (in fact should be right on the borderline). The major reason of preventing a clear acceptance recommendation is that the performance improvement is still marginal in places, like CIFAR-10 ResNet18 architecture. The standard deviation well covered the difference between the average classification error.

---

### Official Review · Reviewer_cHJU · 2023-11-01

**Soundness:** 2 fair
**Presentation:** 2 fair
**Contribution:** 2 fair
**Rating:** 6
**Confidence:** 3

**Summary:**

The paper proposes a modification to softmax when using softmax in conjunction with cross-entropy for classification tasks to combat the following problem: when the magnitude of the partial derivative of the softmax with respect to the class outputs decays rapidly, model tends to overfit (but converges faster). On the other hand, if the magnitude of the partials decays too slowly, the model takes a longer time to converge (but tends to generalize better). The modification introduces a hyperparameter $\beta$ where small $\beta$ encourages rapid decay of the partials while large $\beta$ encourages slow decay of the partials. To combine the best of both worlds, the paper proposes a warm-up scheme by starting with a small $\beta$ so that the model will converge quickly and then increase $\beta$ to discourage overfitting and overconfidence.

**Strengths:**

Paper proposes a simple modification to softmax in conjunction with a warm up scheme with respect to the margin parameter $\beta$ to get faster convergence and better generalization.

**Weaknesses:**

The warm-up scheme does not seem to provide a significant advantage over prior proposed modifications to softmax (e.g A-softmax) or does worse according to table 2 in the paper.

**Questions:**

What does the training loss look like across epochs for the warm-up schedule (more specifically could you plot another curve in figure 5 displaying the loss over epochs for the warm-up schedule)?

---

> ### Author Response · Authors · 2023-11-17
> **Response to Reviewer cHJU**
>
> We appreciate the reviewer for taking the time to review our paper.
>
> * Firstly, in response to the reviewer's queries, we have added supplementary information regarding the warm-up training loss curve in Fig. 5.
>
> * Secondly, considering review’s concerns, we would like to provide additional explanations about the significance of our work.
>
> As mentioned by the reviewer in “Weaknesses”, the warm-up scheme shows an improvement over the previously proposed adjustment for Softmax, but the improvement is not significant. Addressing review’s concerns about the performance improvement, we repeated the results more times and presented the mean and standard deviation of the results in Tab.2 and Tab.3 to support our conclusions. Furthermore, considering that these two approaches address the problem from different perspectives, where A-Softmax statically considers the margin in the output space, while the warm-up scheme dynamically adjusts gradient decay coefficients during the optimization process. Therefore, we believe that the presentation of such results can provide some new insights.
>
> On the other hand, our paper also focuses on the uncertainty estimation of the classification results, i.e., whether the model probability outputs can reflect the accuracy prediction in classification. Perfect calibration of neural network can be realized when the confidence score reflects the real probability that the classification is classified correctly. Formally, the perfectly calibrated network satisfied ${\rm P}\left( {\hat y = y|\hat p = p} \right) = p$ for all $p \in \left[ {0,1} \right]$. However, probability outputs of modern model display unwarranted over-confidence. Our work reveals that a probability-dependent gradient decay rate is a factor contributing to the over-confidence of modern models in classification. The optimization with a small gradient decay rate presents a more stringent curriculum learning for samples, meaning that challenging samples only gain confidence after easy samples have been sufficiently trained to a certain level of confidence. This leads to different samples having greater distinctiveness in the optimization. Correspondingly, our presented analysis, detailed in Section 3, along with more experimental results shown in Fig. 6, Table 3, and Appendix, demonstrates that a substantial increase in the model-calibrated criterion Expected Calibration Error (ECE) can be achieved when employing large probability-dependent gradient decay in modern model. To better present the work in model calibration, we have modified presentation in section 1 and section 2 to emphasize this point.
>
> In addition to methodological innovations, our work can provide some insights and understanding, especially concerning concepts such as model calibration, large-margin Softmax, probability-dependent gradient decay, curriculum learning, and related topics in deep learning optimization.

---

> > ### Author Response · Authors · 2023-11-17
> > **Response to Reviewer cHJU**
> >
> > Dear Reviewer cHJU,
> >
> > Thank you for your valuable feedback on our submission. We have read your comments carefully and have addressed them in our rebuttal. We would be grateful if you could acknowledge if our responses have addressed your comments. Thank you again for your time and consideration.
> >
> > Best regards,
> >
> > Authors of paper 1676.

---

### Meta-Review · Area_Chair_LJwh · 2023-12-13

**Metareview:**

The authors introduce a new parametrization of the softmax through a "gradient decay" hyperparameter. They use the parameter to explain connections of this parameter with curriculum learning sequence, local Lipschitz constraint and model calibration. Additionally, they propose a new warm up methodology to dynamically adjust the gradient decay parameter. The main concern over the paper is around the significance of the results. The reviewers note that warmup strategy proposed in the paper based on the interpretations of the results does not yield substantial boost in performance. The authors claim that the warmup algorithm is not the main crux of the paper, but rather an add-on to their interpretations of how their gradient decay parameter captures various aspects of learning. The metareviewer does not find this reasoning convincing. If the main contribution is the mechanistic understanding, the empirical evidence has to be overwhelming and unconvincing of the importance. Yet most of the interpretations of the implications of gradient decay parameter are based on one 2 layer FCNN trained only on MNIST, which is not very satisfactory. While it might not be possible to train state of the art vision models, the kind of empirical analysis in this paper can be easily made over many vision datasets and model architectures on small-medium scale to show convincing evidence.

**Justification For Why Not Higher Score:**

As noted in my meta review, the paper overall lacks significant convincing results. The empirical results are not fully convincing of the claims made.

**Justification For Why Not Lower Score:**

NA

---

### Decision · Program_Chairs · 2024-01-16

Reject